Resource

# A panel dataset of COVID-19 vaccination policies in 185 countries

Emily Cameron-Blake[1,2,5]✉, Helen Tatlow[1,5], Bernardo Andretti[1,3], Thomas Boby[1], Kaitlyn Green[1], Thomas Hale[1], Anna Petherick[1], Toby Phillips[1], Annalena Pott[1], Adam Wade[1] & Hao Zha[1,4]

We present a panel dataset of COVID-19 vaccine policies, with data from 01 January 2020 for 185 countries and a number of subnational jurisdictions, reporting on vaccination prioritization plans, eligibility and availability, cost to the individual and mandatory vaccination policies. For each of these indicators, we recorded who is targeted by a policy using 52 standardized categories. These indicators document a detailed picture of the unprecedented scale of international COVID-19 vaccination rollout and strategy, indicating which countries prioritized and vaccinated which groups, when and in what order. We highlight key descriptive findings from these data to demonstrate uses for the data and to encourage researchers and policymakers in future research and vaccination planning. Numerous patterns and trends begin to emerge. For example: 'eliminator' countries (those that aimed to prevent virus entry into the country and community transmission) tended to prioritize border workers and economic sectors, while 'mitigator' countries (those that aimed to reduce the impact of community transmission) tended to prioritize the elderly and healthcare sectors for the first COVID-19 vaccinations; high-income countries published prioritization plans and began vaccinations earlier than low- and middle-income countries. Fifty-five countries were found to have implemented at least one policy of mandatory vaccination. We also demonstrate the value of combining this data with vaccination uptake rates, vaccine supply and demand data, and with further COVID-19 epidemiological data.

The COVID-19 pandemic drove an unprecedented effort to develop and deploy vaccines at scale. As of 15 June 2022, 37 COVID-19 vaccines were in Phase 3 clinical trials, authorized for early use or approved for full use (https://www.nytimes.com/interactive/2020/science/coronavirus-vaccine-tracker.html), with some available for use in infants as young as 6 months old[1,2]. Eleven vaccines have now been given Emergency Use Listing (EUL) from the World Health Organization (WHO)[3], and more than 11.9 billion (as of 15 June 2022, https://ourworldindata.org/covid-vaccinations?country=OWID_WRL) COVID-19 vaccine doses have been delivered worldwide[4–6]. Nonetheless, more than 30% of the world population remains unvaccinated[3], and COVID-19 vaccines and public health measures to promote them have generated substantial political controversy in many countries[7].

Given that vaccines against COVID-19 and other diseases will continue to play a critical role in combating the COVID-19 pandemic and future diseases, there is an enormous need to better understand both

[1]Blavatnik School of Government, University of Oxford, Oxford, UK. [2]School of Social and Political Science, University of Edinburgh, Edinburgh, UK. [3]Brazilian School of Public and Business Administration, Getulio Vargas Foundation, Rio de Janeiro, Brazil. [4]Bartlett School of Sustainable Construction, University College London, London, UK. [5]These authors contributed equally: Emily Cameron-Blake, Helen Tatlow. ✉e-mail: E.A.Cameron-Blake@sms.ed.ac.uk

**Table 1 | OxCGRT vaccination policy indicators**

| Indicator | Description | Measurement | Coding |
|---|---|---|---|
| V1 - vaccine prioritization | Records the ranked position for 52 different categories within a country prioritization plan | Rank order | Blank – category not selected for prioritization<br>1, 2, 3, 4... – category has been selected for prioritization; number represents the rank of prioritization; equal-ranked categories will share the same number |
| V2 - vaccine eligibility/ availability | Records the categories, regardless of their position in a prioritized rollout plan, that are receiving vaccines | Categorical/ binary | Blank – no data<br>0 - vaccines are not being made available to this category<br>1 - vaccines are being made available to this category |
| V3 - vaccine financial support | Records how vaccines are funded for each category identified in V2 as receiving vaccines. | Ordinal scale | Blank - no data<br>1 - full cost borne by the individual (or through private health insurance) or no policy<br>2 - partially funded by government and individual pays nominal fee<br>3 - fully covered by government funding and thus free at the point of use |
| V4 - vaccine requirement/ mandate | Reports the existence of a requirement to be vaccinated for each category | Binary | Blank - no data<br>0 - no requirement to be vaccinated<br>1 - requirement to be vaccinated |

The construction of these measures is described more fully in Methods or in further detail in our open-source data repository on GitHub. For more detail, see Supplementary Table 1.

how governments use vaccines and the factors that drive or inhibit uptake. Within the latter, the policies and measures that governments adopt to promote vaccination play an important role. This resource article introduces a new dataset from the Oxford COVID-19 Government Response Tracker (OxCGRT) project[8] that provides cross-national and longitudinal information on COVID-19 vaccination policies across the world. The new dataset covers vaccine prioritization policies (V1), vaccine availability and eligibility by group (V2), whether there is a cost to individual recipients at the point of use (V3) and mandate policies (V4) (Table 1). Developed in consultation with governments, policymakers, stakeholders in the field and the OxCGRT expert advisory board, this dataset allows researchers and policymakers to systematically compare government vaccination strategies and assess distributional capabilities (see Methods for more information on dataset design). Through combination with other datasets, further analyses, such as on the impact of distribution policies on uptake, are made possible.

The dataset is composed of four indicators that report national-level COVID-19 vaccination policies for 52 population categories from 1 January 2020 to 31 December 2022 for 185 jurisdictions in the national database, as well as subnational jurisdictions (provinces) for Canada and (states) for the United States (subnational-jurisdiction data for other countries are in development and will be added to the online data repository as they become available). Collected by a specially trained team of OxCGRT volunteer data contributors, the dataset has been reviewed for quality assurance and accuracy by core OxCGRT researchers and a specially trained review team. The results presented here focus only on the national-level data. The 52 population categories include general and at-risk age groups (in 5-yr intervals for individuals over 16 yr), occupational categories (based on exposure and transmission risk and economic function—groups of people working in occupations critical to the economic function of the country, including essential workers and airport/border staff) and medical categories (based on vulnerability) (Table 2). These closely reflect the categories listed in the WHO Strategic Advisory Group of Experts on Immunization (SAGE) values framework for the allocation and prioritization of COVID-19 vaccination, a framework that guides the allocation of vaccinations under limited supply to specific population groups to meet specific outcomes.

Here we present some variations and temporal patterns of COVID-19 vaccine prioritization, eligibility, distribution, cost and mandates by group, with the aim of illustrating how the new dataset can offer insights for researchers and policymakers. By mapping variation in vaccination policies over time, this dataset provides a

**Table 2 | OxCGRT vaccination policy categories**

| Categories |
|---|
| **General age categories** |
| • 0–4 yr infants<br>• 5–15 yr young people<br>• 16–80+ yr (listed separately in 5-yr groupings) |
| **Vulnerable groups** |
| • Clinically vulnerable/chronic illness/significant underlying health condition (excluding elderly and disabled)<br>• Residents in an elderly care home<br>• People living with a vulnerable/shielding person or other priority group<br>• Disabled people<br>• Pregnant people<br>• At-risk 16–80+ yr (listed separately in 5-yr groupings) |
| **Economic function** |
| • Frontline retail workers<br>• Other 'high-contact' professions/groups (taxi drivers, security guards)<br>• Airport/border staff<br>• Factory workers<br>• Frontline/essential workers (when subcategories not specified) |
| **Education** |
| • Educators<br>• Primary and secondary school students<br>• Tertiary education students |
| **Healthcare workers** |
| • Healthcare workers/carers (excluding care home staff)<br>• Staff working in elderly care homes |
| **Public function** |
| • Government officials<br>• Military<br>• Police/first responders<br>• Religious/spiritual leaders |
| **Socially vulnerable** |
| • Ethnic minorities<br>• Refugees/migrants<br>• Crowded/communal living conditions (dormitories for migrant workers, temporary accommodations) |

The construction of these measures is described more fully in Methods or in further detail on our open-source data repository on GitHub. For more detail, see Supplementary Table 2. Alternate proposals for vaccine distribution have also been suggested, notably 'The Fair Priority Model' that promotes a more ethical distribution of vaccines, ensuring that hoarding of vaccines and vaccine waste are reduced[9].

**Table 3 | Groups prioritized in official published plans for COVID-19 vaccination in the first round of vaccination rollout by eliminator and mitigator countries**

| | Economic function | | | Healthcare | | | Vulnerable Groups | | | | |
| --- | --- | --- | --- | --- | --- | --- | --- | --- | --- | --- | --- |
| | Airport/border staff | Frontline/essential workers | Frontline retail | Elderly care home staff | Healthcare workers | Residents in elderly care homes | Clinically vulnerable/chronic illness/significant underlying health condition | People living with a vulnerable/shielding person or other priority group | Disabled people | Pregnant people | People at risk |
| **Eliminators** | | | | | | | | | | | |
| Australia | ✓ | | | ✓ | ✓ | ✓ | | | | | |
| New Zealand | ✓ | ✓ | | | ✓ | | | | | | |
| China | ✓ | ✓ | | | ✓ | | | | | | |
| South Korea | | | | ✓ | ✓ | ✓ | | | | | |
| Fiji | | ✓ | | | | | | | | | |
| Singapore | | ✓ | ✓ | | ✓ | | | | | | |
| Tonga | ✓ | ✓ | | | ✓ | | | | | | |
| Iceland | | | | ✓ | ✓ | ✓ | | | | | |
| Solomon Islands | ✓ | ✓ | | | ✓ | | | | | | |
| **Mitigators** | | | | | | | | | | | |
| Brazil | | | | ✓ | ✓ | | | | | | |
| Canada | | | | ✓ | ✓ | ✓ | | | | | ✓ |
| Switzerland | | | | ✓ | ✓ | ✓ | ✓ | ✓ | | | |
| Spain | | | | ✓ | | ✓ | | | | | |
| France | | | | ✓ | | ✓ | | | | | |
| Italy | | | | | ✓ | | | | | | |
| Turkey | | | | | ✓ | | | | | | |
| United States | | | | ✓ | ✓ | ✓ | | | | | |
| Morocco | | | | | ✓ | | | | | | |

means of responding to critical questions about the role of vaccines in the COVID-19 pandemic. The four indicators are presented in the dataset alongside vaccination rates so that users can easily generate cross-national and within-country comparisons to explore associations between vaccination policies, those prioritized for vaccination and vaccine uptake. The dataset's jurisdiction–day format purposefully facilitates merging with other data sources. For example, combining these four vaccine policy indicators with data about other kinds of pandemic policies—relating to closures, economic support and health, such as testing—allows for assessment of the interplay of vaccination programmes within the broader context of pandemic management. Combining these data with other social science indicators broadens the possibilities further, towards considerations of the impacts of vaccination policies in different economic, political, cultural or religious contexts, and how controversies towards COVID-19 vaccines may have shaped policy. Comparisons between countries with similar demographics, or those with unique demographics and geography might highlight best practice for future pandemics[10].

## Results

To motivate use of this new dataset, in this section we illustrate some initial observations using the four vaccine policy indicators. First, we focus on the first two indicators (V1 and V2). We note common patterns in vaccination prioritization strategies according to the broader strategies guiding countries' pandemic management, and by country income level. We then observe country-level variation in both prioritization and actual distribution by age group and other kinds of vulnerability

categories. Second, we consider the additional two indicators (V3 and V4), briefly discussing the cost of vaccination to individuals, before exploring vaccine mandate policies. Although we find limited variation relating to cost across countries—almost all countries provided the vaccine to people free of charge—we detail the diversity of countries' vaccine mandate policies, ranging from non-existent, to policies specific to certain age groups, occupation groups and populations carrying out specific religious activities, or working in named locations.

### Comparing countries' prioritization plans

The vaccine prioritization (V1) indicator captures the order in which governments planned to deploy COVID-19 vaccines to different categories of people. It depicts the published plans of all countries in the OxCGRT dataset (only Eritrea, of 185 countries, has yet to begin COVID-19 vaccinations or publish a vaccination plan). These plans were primarily used to ration limited supplies at a time when global vaccine supply was constrained. As such, they help reveal the evolving priorities governments held for vaccine deployment.

Broadly, we can distinguish two ideal typical COVID-19 strategies: elimination and mitigation. Roughly, studies have categorized these strategies by observing how eliminators aimed to keep the virus out of the country and to prevent community transmission. In contrast, mitigators sought to reduce the impact of community transmission by flattening the curve and reducing cases and deaths[11–14]. Many countries in the WHO West Pacific Region adopted policies closer to the elimination model, perhaps in part due to the region's previous experience with SARS[11,13]. Relatedly, it has also been hypothesized that island

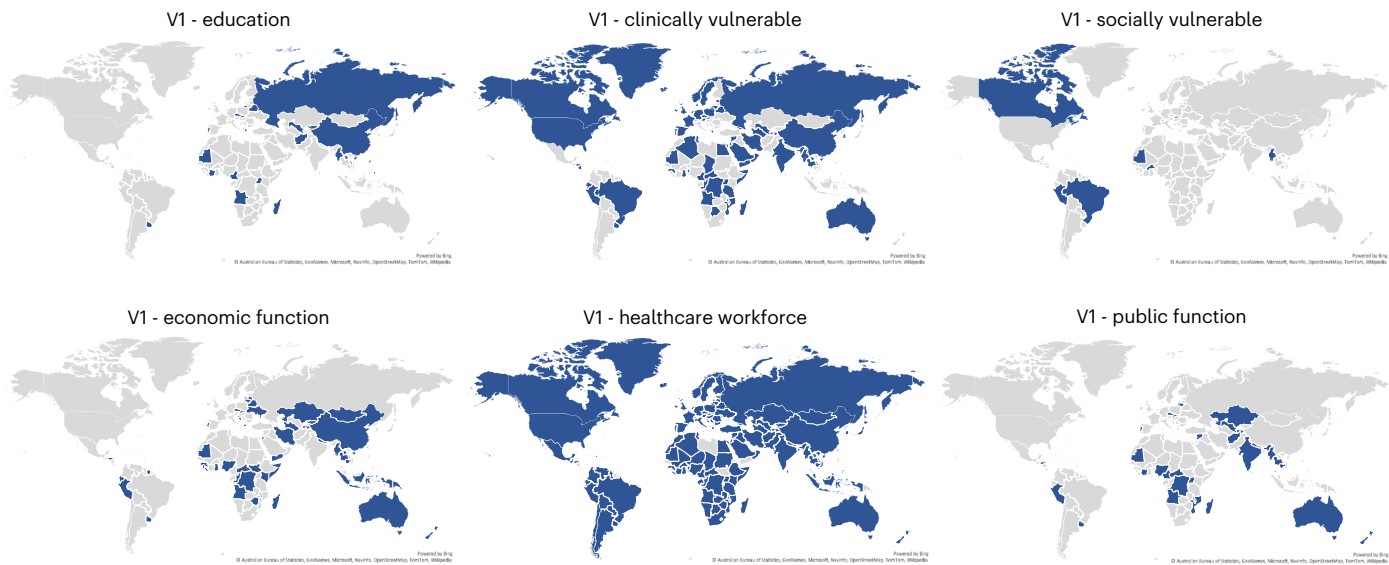

**Fig. 1 | Vaccine prioritization themes by country.** Countries in blue prioritized certain aspects or functions of population groups as part of their first round of COVID-19 vaccinations (position rank 1). Education (educators, primary/tertiary students, tertiary education students); Clinically vulnerable (clinically vulnerable/chronic illness/significant underlying health condition); Socially vulnerable (ethnic minorities, refugees/migrants, crowded/communal living); Economic function (frontline retail workers, frontline/essential workers, airport/border staff, other high-contact professions, factory workers); Healthcare workforce (healthcare workers, staff in elderly care homes, people living with a vulnerable person); Public function (government officials, police/first responders, military, religious/spiritual leaders).

nations may have been more likely to pursue elimination models during COVID-19 due to their geographical advantage of isolation and ability to seal national borders, although for many low-income countries (LICs) and LMICs, this came at appreciable economic cost[11,15–18]. Finally, early responses to the COVID-19 pandemic have also highlighted distinctions between mitigation and elimination strategies based on the Organisation for Economic Co-operation and Development (OECD) countries, such as high-performing surveillance systems with early and targeted contact tracing and widespread efficient testing in eliminator countries[12,14].

Interestingly, we observe a clear distinction in the vaccination rollout strategies of countries that have been characterized as pursuing an elimination strategy throughout much of the COVID-19 pandemic, compared to countries that pursued a mitigation strategy[11–14,19]. Table 3 highlights how nine selected eliminators (selected as examples from a comprehensive list of eliminators, see Supplementary Table 3), such as Australia and New Zealand, prioritized frontline workers and border staff alongside clinically vulnerable people (those with co-morbidities and/or predispositions to illness) in their first rounds of vaccination (that is, positioned rank 1), while nine selected mitigators (see Supplementary Table 3 for comprehensive list of mitigators) focused instead on vaccinating only clinically vulnerable and elderly populations, along with healthcare workers. Looking across all countries in our dataset, while 11 out of 19 (58%) Western Pacific countries prioritized first-round vaccinations on the basis of economic function, only 42 out of 164 (26%) non-Western Pacific countries did so. When looking at the OECD subset, we observe the same pattern: 2 out of 5 countries that adopted elimination in the early stages of the pandemic (40%) prioritized vaccination on the basis of economic function versus 4 out of 32 (12%) mitigators. Finally, the same descriptive pattern holds for island nations (16 out of 32, 50%) when contrasted to non-island nations, with 16 (50%) island countries adopting economic function prioritized vaccination versus non-island countries (38 out of 151, 25%). In contrast, only 3 Western Pacific countries (16%) prioritized clinically vulnerable people, versus 80 (49%) for countries outside the region. However, we observe no differences in descriptive patterns for either OECD or island country groupings. While 3 out of 5 OECD countries

that adopted elimination strategies in the early stages of the pandemic (60%) prioritized clinically vulnerable people, 19 out of 32 mitigators did so. Finally, 44% of island (14 out of 32) and 46% of non-island countries (70 out of 151) prioritized clinically vulnerable people.

When we map country differences in the most- (or equal-most-) prioritized categories of people (Fig. 1), we observe that almost all countries in our dataset placed healthcare workers among the first to receive COVID-19 vaccines (91% of countries, or 168/185) presumably both to reduce transmission and mitigate health impacts for those at high risk of exposure. Following healthcare workers, 54% (101) of countries also prioritized populations deemed to be clinically vulnerable. Seventy-four countries (40%) prioritized groups of people critical to the economic function of the country, 45 prioritized (24%) categories related to the public function of the country, and 27 countries (15%) focused on educators and/or students for COVID-19 vaccinations.

**Comparing countries' plans with actual distribution**

By the end of 2021, 62 (34%) countries had published COVID-19 vaccination rollout plans and prioritization lists, often with rough estimates for the timing of each phase. Of those, 48 countries began vaccination before the end of 2021 (Fig. 2), and by 31 March 2021, 107 more countries had published their COVID-19 vaccination plans and priorities. In countries with access to vaccines, deployment swiftly followed plans (captured by indicator V2, which records the groups actually receiving a vaccine in a country). Because vaccine access was highly related to countries' income levels[20], Fig. 2 groups countries according to the World Bank income classification groups for 2021 (https://datatopics.worldbank.org/world-development-indicators/the-world-by-income-and-region.html). HICs were the first to publish vaccination plans to secure COVID-19 vaccines and to begin administration of vaccines for their populations, reflecting their greater ability to produce and purchase vaccines. Figure 2 shows that the publication of plans and subsequent administration of vaccine doses then followed in upper middle-income countries (UMICs), LMICs and then LICs, respectively. HICs moved quicker to expand vaccination eligibility to the 12+ yr age group than LMICs. It also shows that despite publishing plans and beginning vaccinations at later dates than HICs and UMICs,

Vaccine rollout across countries

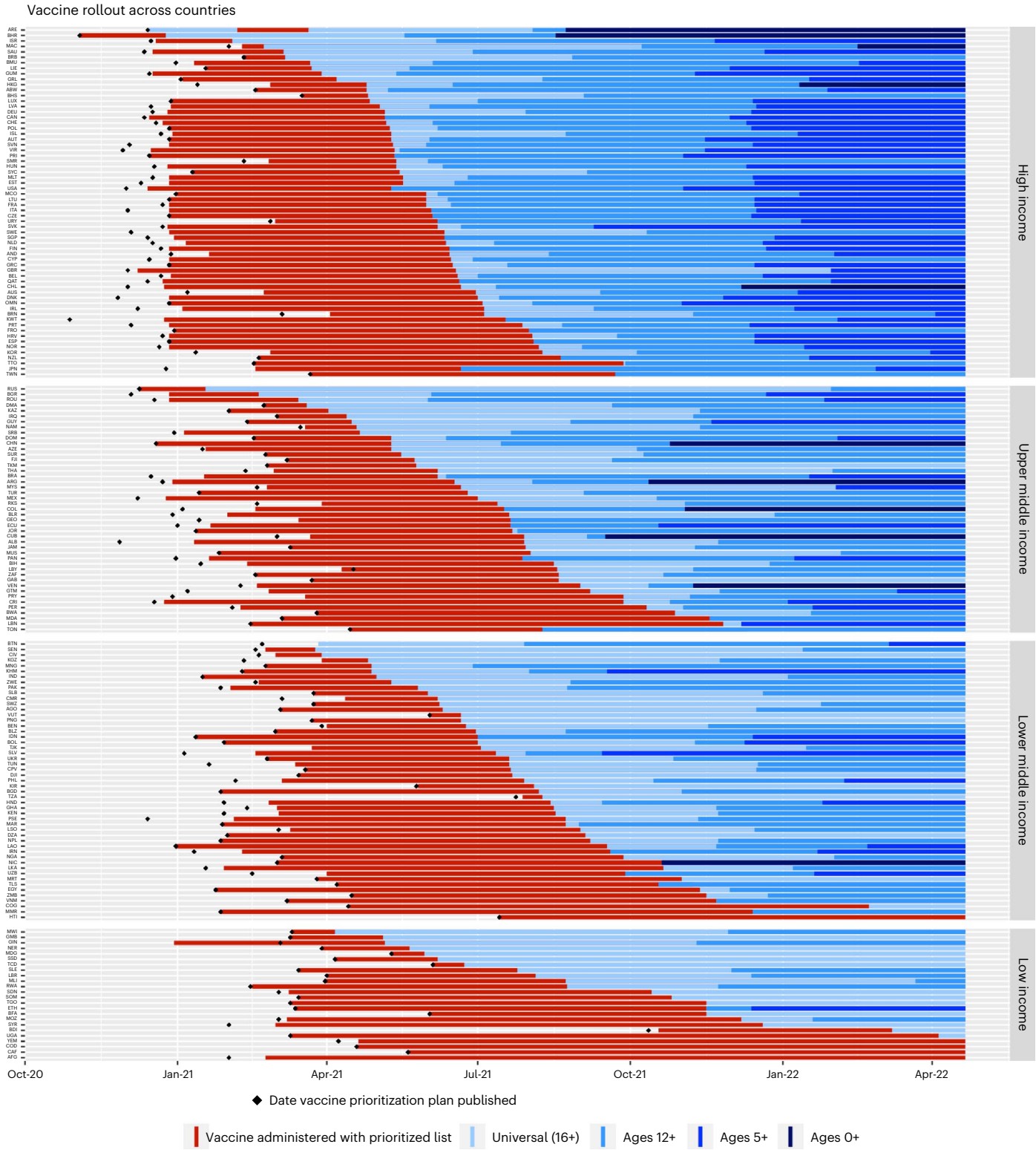

**Fig. 2 | Variation in timing of vaccine rollout.** The timing of COVID-19 vaccination plans and administration/eligibility for vaccines varied greatly between countries and territories. Data current until 15 July 2022.

some LMICs and LICs quickly moved from prioritized vaccinations by age or occupation to universal access (defined in our dataset as being available widely to all those over the age of 16+ yr or 18+ yr, with age floor dependent on vaccine brand). LMICs and LICs have had far fewer vaccine doses available to them than HICs and UMICs (https://data. undp.org/vaccine-equity/accessibility/), so one possible explanation

for the swift movement towards universal access in LICs and LMICs may be an attempt to vaccinate larger proportions of their populations in the face of vaccine hesitancy. However, there are conflicting views on whether LMICs and LICs are more or less vaccine hesitant towards COVID-19 vaccines than HICs and UMICs[21–23]. Another possible explanation for the quick move to (ostensibly) universal access (despite

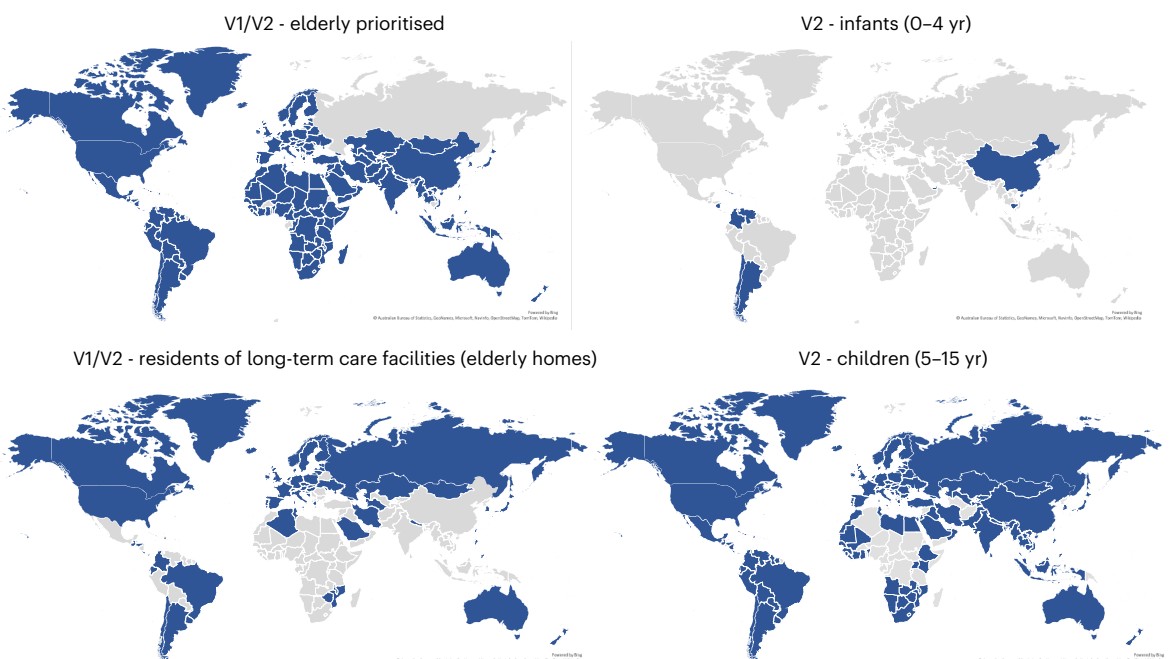

**Fig. 3 | Vaccination of elderly populations, children and infants.** Variation in prioritization and eligibility (V1 and V2) of elderly people and LTCFs (elderly homes) for COVID-19 vaccination before universal eligibility (16/18+ yr vaccine dependent) for COVID-19 vaccination. Following 'universal' COVID-19 vaccination eligibility (V1) in most countries, eligibility (V2) for infants (0–4 yr) and children (5–15 yr) were added at later dates. Data current to 15 June 2022.

low vaccine supply) may have been to reduce public perception of favouring or excluding certain groups in the face of a public health emergency and with varying social and geographic challenges in the distribution of vaccines[24,25].

## Comparing categories by age and vulnerability status

Only 11 countries in our dataset did not specifically prioritize elderly populations (the age cut-off for 'elderly' is defined locally, see Methods) in their plans and rollouts of COVID-19 vaccines (Bangladesh, Burkina Faso, Cambodia, Macau, Namibia, Russia, Trinidad and Tobago, Uganda, Uruguay, Zimbabwe and Taiwan). Long-term care facilities (LTCFs; nursing homes, elderly care homes and so on) are a concept and cultural phenomenon predominantly found in some relatively wealthy countries[26]. This is reflected in our data (Fig. 3), with most African, South Asian and Middle East countries not prioritizing LTCFs in their plans and rollouts. In contrast, while not prioritizing its elderly population in general, Russia did prioritize vaccinations at LTCFs for elderly people before moving to universal eligibility.

At the younger end of the age scale, children and infants were not included in initial COVID-19 prioritization plans and indeed, the WHO stated that children were a lower priority group as they are less likely to experience serious symptoms[27]. Clinical research into the safety and efficacy of vaccines in children came later[28]. Because clinical trials split children into groups of young children (~5–11 yr) and adolescents (~12–17 yr)[29], many countries' regulatory agencies approved COVID-19 vaccines for children in similar age groups at around the same time as research emerged and as vaccine supplies allowed (Fig. 2). Children 12 and above first became eligible for COVID-19 vaccinations on 6 May 2021 in the territory of Nunavut, Canada, and Bahrain made the COVID-19 vaccine available to children aged 3+ yr on 17 August 2021. As of 15 June 2022, only 12 countries were vaccinating infants (children aged 0–4 yr), 6 being Central and South American countries (Fig. 3), and 26 countries in our dataset have not been recorded as vaccinating children or infants (ages 0–15 yr).

Countries prioritized many professional categories, some of which (for example, healthcare workers) were gendered to varying extents

in different countries. Pregnant people were first made eligible for vaccinations in 2021. They were not initially prioritized for COVID-19 vaccination due to a lack of clinical trial data on the effect on unborn children and pregnant people. From January 2020 to 15 June 2022, 21 countries explicitly prioritized pregnant people in their official vaccination rollout plans (V1) (Fig. 4), and 54 out of 185 countries reported specifically administering vaccinations to pregnant people during their vaccination rollouts (V2).

Similarly, strikingly few countries have prioritized refugees, migrants and ethnic minorities in their COVID-19 vaccination plans or actual rollouts (Fig. 4), despite reports of disparities in the risk of COVID-19 infections and in health outcomes among these groups[30–34]. Australia, Brazil, Canada and New Zealand all prioritized Indigenous populations in their plans and administration of COVID-19 vaccines (Aboriginal and Torres Strait Islanders in Australia, Indigenous People, traditional riverine and quilombola communities in Brazil, First Nations in Canada, Māori and Pacific People and their extended families/whanau that care for them in New Zealand). In early June 2021, Greece began a vaccination programme specifically for refugees starting with the island refugee camps, then moving to the mainland camps. Some African countries with high rates of internal migration have prioritized the vaccination of migrants and 'travellers' (a term sometimes used for migrants) (Fig. 4) (https://africacenter.org/spotlight/african-migration-trends-to-watch-in-2022/). Many other countries may have made COVID-19 vaccines available to refugees and migrants but may not have explicitly noted this in official documentation.

## Vaccine cost and mandates

The dataset reveals that nearly all countries provided the COVID-19 vaccine free of charge to individuals at the point of delivery (V3). From data freely available online from government sources, we observed that Botswana, India, Pakistan and Turkmenistan required a small fee for COVID-19 vaccination from individuals, in some circumstances. In Taiwan, vaccines were free for those meeting eligibility criteria, but individuals could self-fund vaccination in advance of becoming

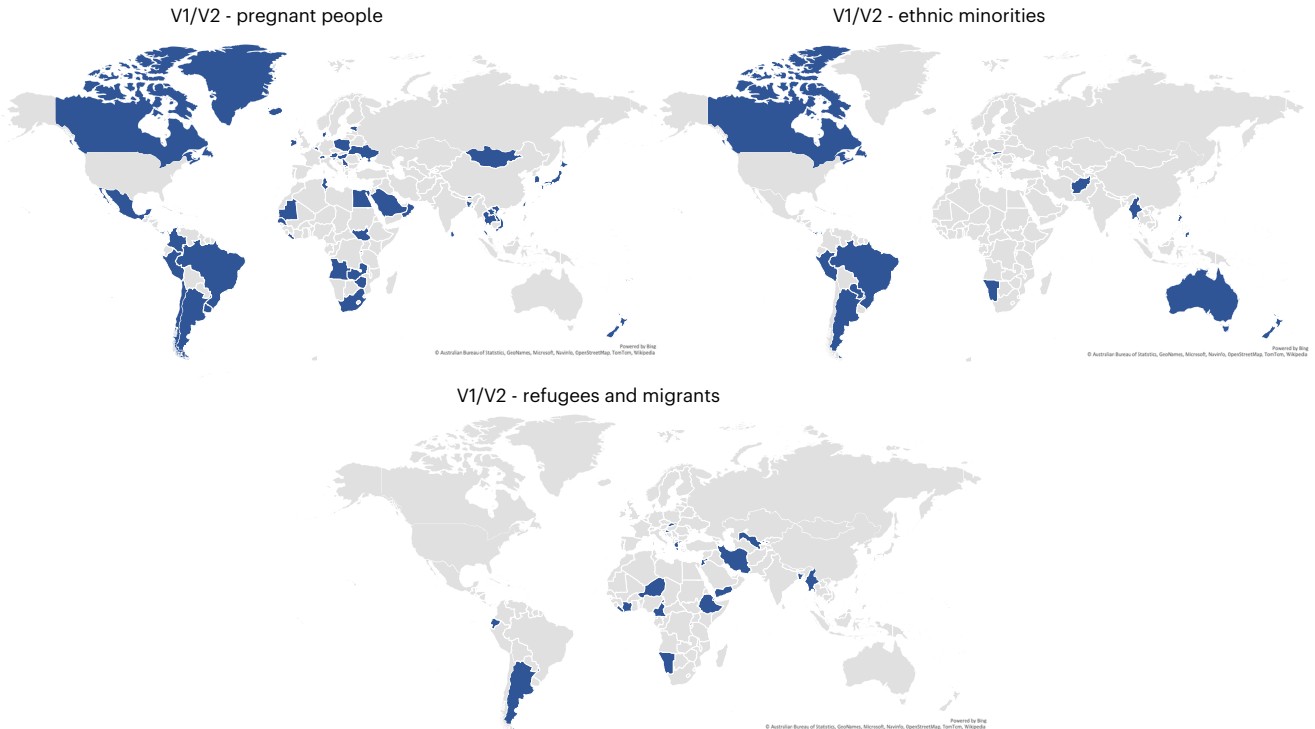

**Fig. 4 | Vaccination of pregnant people, ethnic minorities and refugees/migrants.** Some countries prioritized certain groups in their published plans (V1), while other countries added eligibility for these groups ad hoc as vaccinations were already underway (V2). This image shows which countries either specifically prioritized or added eligibility for these groups. Data current to 15 June 2022.

eligible for free vaccines. Some countries enabled individuals to acquire the vaccination privately if they wished or encouraged vaccine tourism in efforts to boost tourism during the pandemic. In late 2021, Russian tourists were accessing free non-Sputnik vaccines in Croatia via specially arranged tours[35]. Not long after, Russia began to offer their Sputnik COVID-19 vaccine at a cost to tourists (although visas to visit the country were an issue for some)[36]. Cities and states of the United States, Indonesia, Cuba, the Maldives and the UAE began to use free COVID-19 vaccinations to increase tourism, and travel agencies in the United Kingdom, India and the United States (among others) offered package deals for individuals not yet eligible in their own country to access the vaccine in another[37–39]. These cases are not recorded in our V3 ordinal indicator data but are captured in qualitative coding notes within the dataset.

We define mandatory COVID-19 vaccination as a government policy requirement to be vaccinated against COVID-19 to work in a specific occupation, or for a specific age group of citizens to be vaccinated. We see these policies as distinct from COVID-19 passports or certification to gain entry to non-essential services or for international travel based on demonstrating immunity or negative test results. These 'passport' style policies are defined in the OxCGRT database as 'differentiated policies' and are part of the OxCGRT Non-Pharmaceutical Intervention (NPI) dataset[8,40]. For 10 of these original indicators, we report two policies, one for vaccinated and another for non-vaccinated individuals, viewing requirements to present passes as a de facto closure (see online documentation and Methods for further detail).

COVID-19 vaccine mandates (V4) were first introduced in Jakarta, Indonesia, in February 2021 for all adults[41]. As of 15 June 2022, 55 countries (29%) of our dataset have currently, or had at one time, at least one vaccine mandate policy. Figure 5 shows that mandatory vaccination policies accelerated in implementation around July 2021, commonly according to occupation (83%), with age-based mandates being far less widespread (17%).

The most common groups to have been mandated for COVID-19 vaccination were related to occupation: government officials (36), healthcare workers (29), staff at LTCFs (15) and educators (22). Hence, in most countries, mandates only affected a small and specific proportion of the population, as opposed to COVID-19 passports which affected the whole population to control their access to services. Six countries (Germany, Ethiopia, Kuwait, New Zealand, Russia and Tonga) mandated vaccines for socially vulnerable and or other vulnerable populations (see Table 2). Nine countries, including Indonesia, Tajikistan and Turkmenistan mandated COVID-19 vaccination for the entire adult population. Ecuador mandated vaccination for all those over 5 yr and in Costa Rica, all those aged 3–18 yr must be vaccinated ("×" symbol in Fig. 5). The qualitative notes section of this new vaccine dataset records how some countries introduced specific vaccine mandates for particular groups relevant to their local context, such as those observing the Hajj in Saudi Arabia, Jahh in Bangladesh, and maritime and port workers in Singapore.

## Discussion

Systematic tracking of COVID-19 vaccination policies and their variation is critical to understand and compare the strategies that countries undertook, and to learn lessons for future pandemics and ongoing vaccination needs. However, collecting such data presented certain challenges due to the complexity and specific considerations of each country/government. We review and address some of these limitations in this section.

First, not all vaccination documents and details are publicly accessible, and language translation barriers are common. As such, recorded sources may range from publicly available news sources, and in some instances, the social media accounts of governments. The detailed notes we have published alongside all entries contain archived sources for future reference and for confirmation of policies.

Second, translating and interpreting heterogeneous policies into the broad categories requires contextual judgement on the part of our

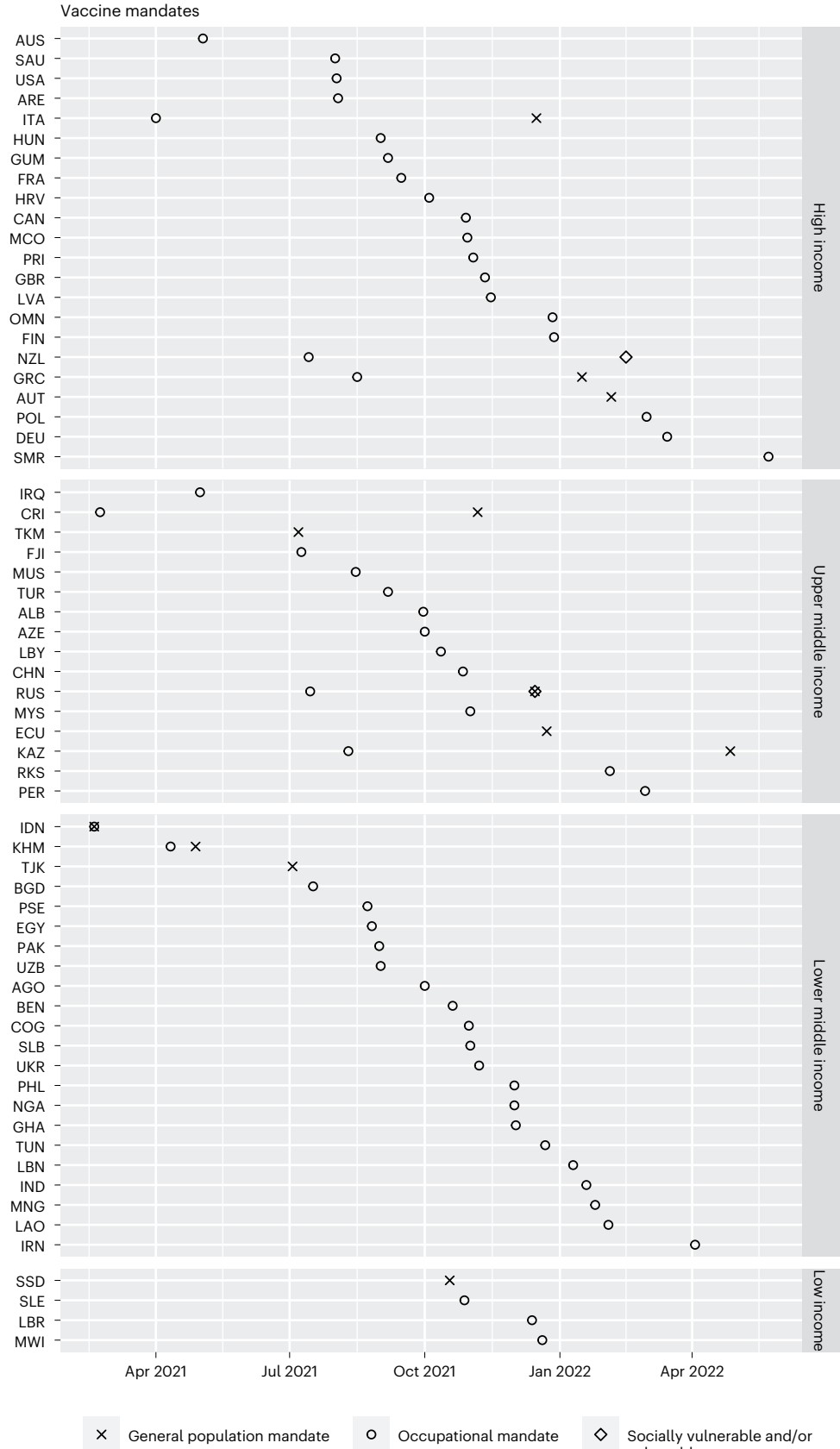

**Fig. 5 | Mandatory vaccine timing and groups.** Introduction of mandatory COVID-19 vaccine (V4) for occupational groups, general population (captured if any age group mandated) and vulnerable groups and/or socially vulnerable groups (refer to Table 2 for specific categories). The nations that introduced mandatory vaccinations are grouped by World Bank income groupings. Data current to 15 June 2022.

specially trained data contributors. As a result, similar policies risk varying interpretations. An initial exercise to determine contributor interpretation variance determined that 90% of entries were consistently interpreted (see Methods).

Third, the 52 categories selected for the V1–V4 indicators (16 general age categories, 14 at-risk age categories and 22 health and occupational categories) are not exhaustive; as such, we publish a 'best fit' table in our interpretation guide to aid the classification of groups of people into our categories (https://github.com/OxCGRT/covid-policy-tracker/blob/master/documentation/interpretation_guide.md) (see also Supplementary Table 4). Referral to the detailed qualitative notes may provide more context for which group has been prioritized/made eligible for COVID-19 vaccination.

Fourth, we record an age range as prioritized and/or eligible and/or mandated (V1/V2/V4) if any age within that range is prioritized, eligible for, or mandated to be vaccinated. For example, if 10-yr-olds are prioritized, we would publish '5–15 yr young people' for V1. This choice demonstrates the trade-off between granularity and excess complexity. Therefore, the presence of the '5–15 yr young people' age range category could mean everyone 5+ yr are prioritized, or only people aged 15 yr. Detailed notes in the dataset indicate more granular age floors.

Finally, some vaccination policies apply only in targeted geographic regions. To deal with this variation, we do not require jurisdiction-wide application to code a policy as existing. Instead, if a category of people were prioritized or made eligible for a COVID-19 vaccination in a targeted region of a jurisdiction, they are recorded in our database for the whole jurisdiction. This means that data recorded may not represent the entire jurisdiction. These decisions are detailed in qualitative notes. The dataset includes subnational vaccine policy data for some of the countries with the greatest subnational variation, such as the United States.

## Methods

In this section we describe the design, structure, collection, publication and review of the panel dataset of COVID-19 vaccine policies, with the jurisdiction–day as the unit of analysis. All OxCGRT data are available on GitHub and are licensed under the Creative Commons Attribution CC BY standard. Data users should refer to the project website for updated documentation and methodologies (https://github.com/OxCGRT/covid-policy-tracker). Excel (v.16), Stata (v.17.0) and R (v.4.1.3) software were used for all analysis and images produced in this resource article.

### Indicators

In late 2020, the OxCGRT team began to consider the COVID-19 vaccine policies and strategies that might be deployed globally. Extensive consultations with government officials, policymakers, experts and stakeholders in the relevant fields, consideration of critical gaps in existing data at the time, and capacity to collect such information resulted in a focus on four priority policies to measure (vaccine indicators V1–V4): who was prioritized for vaccines, who was eligible and receiving vaccines and in what order, who was paying for vaccines, and who was mandated to be vaccinated. Additional indicators for future consideration could include which vaccines were approved/deployed in each country, prioritization for additional/booster vaccinations, vaccine dose timing (manufacturer recommendations vs timeline chosen by country for subsequent doses), government vaccination incentives, vaccination education strategies/programmes and internal vs external funding for vaccines.

Vaccine prioritization (V1) records the ranked position for different groups within a country's prioritization plan. This is the official plan of the official order in which categories are to be vaccinated. This is recorded regardless of whether a country has capacity to distribute and administer vaccines. Vaccination eligibility/availability (V2) is a categorical/binary variable that records which categories of people, regardless of their position in a prioritized rollout plan, are currently

receiving vaccinations. This is recorded as a non-zero value if there is evidence that people in this category are being vaccinated, whether this is happening in a targeted geographical region or nationwide. There must be de facto evidence that this is happening to record a non-zero value in V2. For 'elderly' groups, we used local elderly age definitions using qualitative notes where available. In the absence of local guidance on elderly age range, a default of 65+ yr was used. Vaccine financial support (V3) is recorded on an ordinal scale. It reports how vaccines are funded for each category selected in V2 as currently receiving vaccination (1 for full cost to individual, 2 for partially funded by government, 3 for fully funded by government/free). Mandatory vaccination (V4) is a binary variable which reports the existence of a requirement for a category of people to be vaccinated. These are government policy requirements to be vaccinated against COVID-19 to work in a specific occupation, or for a specific group to be vaccinated. This is a mandatory vaccination required as part of occupation or citizenship, and we do not record policies that only 'encourage' voluntary vaccination to access non-essential services, facilities or freedoms. For example, if healthcare workers must be vaccinated to attend their place of work, this is recorded in V4. As detailed in our interpretation guidance, if there is a vaccine mandate in place for workers of certain occupations, and non-vaccinated people in this occupation have the option of testing regularly to opt out of vaccination, we still record this as a mandate in V4. If vaccination is mandatory in a subnational region as a result of an official national or local government policy, we record this in V4. We do not report enforcement of vaccine mandates, just the policy.

The vaccination policy dataset also contains summary indicators. These use an ordinal scale to report one number per indicator to offer a succinct summary on the basis of data entered for variables in the main V1–V4 dataset. These are summarized in our codebook (see Supplementary Information). V2, V3 and V4 policies are reported as non-zero values on the date when the policy came into effect, as opposed to the date they were announced. We report the V1 government rollout prioritization plan on the date the policy was published. For the analysis in this paper, the detailed qualitative notes section was used to manually find specific age floors for when different ages became eligible for vaccination, rather than using the age ranges in the quantitative data due to the limitations described above.

In March 2022, the OxCGRT NPI dataset and data structure was altered to reflect the different policies for unvaccinated and vaccinated people (for COVID-19), where passes or proof of either negative testing or vaccination were required to access different areas of public life (https://github.com/OxCGRT/covid-policy-scratchpad/tree/master/differentiated_vaccination_policies). For ten indicators (C1–8, H6 and H8), both the policy values for non-vaccinated and vaccinated people are reported if there are different policies in place. If there is no differentiation, one value is reported. These differentiated values relate only to 'voluntary access' to elements of public life, such as attending large events or entertainment and dining venues, which are controlled by testing, evidence of previous immunity or vaccination status through a form of certification or proof.

It is valuable to record both mandatory vaccine policies and voluntary vaccine pass/passport policies. This differentiated coding is reported in our NPI dataset with 'V' and 'NV' designations (for example, 'C1V_School closing' and C1NV_School closing') and is distinct from the 'V4 - mandatory vaccination' indicator reported in this paper.

### Data collection

We collected and published data on publicly available sources, which are freely available through internet searches. These are found on government websites, policy briefs and reputable news outlets. The best-quality sources are original policy documents and high-quality media outlets. These sources are codified into the OxCGRT data protocol and entered into the database. A detailed qualitative note (written in English) and web-archived source are provided for context.

The V1–V4 data for the national dataset were initially collected by OxCGRT researchers and a small group of experienced and specially trained volunteer data collectors who received specific training. As of 1 January 2022, data collection for V1–V4 was integrated into the core OxCGRT data collection process, which involves an international team of hundreds of volunteer data collectors covering 90+ unique languages. Where language was a barrier, and where English sources could not be found, local speakers from the pool of OxCGRT contributors were consulted, or translation tools such as Google Translate were utilized. The majority of data collectors are/were postgraduate students at universities around the world. Each data collector completed a bespoke online e-learning course, which took around 45 min to complete, covering the V1–V4 indicators, key coding interpretation points and data quality. Every week of data collection was reviewed by a member of the OxCGRT team for quality and accuracy, and was further reviewed by a trained V1–V4 specific volunteer reviewer (also trained via a bespoke reviewer training e-learning course) and confirmed in the database over the medium to longer term. On GitHub, we published both a codebook and an interpretation guide to standardize interpretation and ensure that all coders were making the same logical choices for data entry. We also published a 'best fit' table which records where categories have been substituted when there is a category announced in a policy that is not listed in V1/2, ensuring the standardization of 'best fit' interpretation (Supplementary Table 4). This highlights the importance of the 'notes' function for each indicator, enabling an in-depth qualitative analysis of the data beyond the numerical summaries. Qualitative notes in our comma-separated values files (CSVs) can be used to find archived original policy documents and specific groups prioritized in each country if a best fit has been used.

## Data publication and summary indicators

The data are published in a standalone 'Vaccines_full.csv' on GitHub, which includes all the 52 categories for V1–V4. They are also summarized in 9 summary indicators, which report one single number for V1 (vaccination prioritization), V2 (vaccination eligibility/availability) categories (overall summary, general population age floor, at-risk age floor, medically clinically vulnerable (non-elderly), education, frontline workers (non-healthcare), frontline workers (healthcare)) and V3. These summary indicators are included in most of our data CSVs on GitHub, including the NPI dataset 'OxCGRT_nat_latest.csv'. The data are published in real time and updated each week by the team of international data collectors.

## Independent data quality check

We conducted a data quality and accuracy check, where two core team researchers checked the data and original source material for 100 randomly selected data entry points, focusing on the period of time when the majority of new policies and policy changes were being made, and during which the majority of data were entered for all four vaccination policy indicators. The researchers independently verified that 90% of the original data entries in the database were correct, by analysing original source material and checking this alongside the database entries. The remaining 10 entries were corrected. Specifics such as date of announcement vs date of policy effect, accuracy of selected groups and ages, and ensuring that any recorded mandates were in fact government mandates were among the various checks in this independent data quality review process.

As part of the weekly data update cycle, a small team of specially trained reviewers are tasked with checking and confirming data entered. As part of this review process, changes to the data are made retroactively, meaning values on past dates may change. As a result, we recommend downloading the most recent data and stating the download date when using them for analysis. The data used for this paper up to 15 June 2022 have been checked thoroughly by the authors. On the rare occasion where data could not be verified or confirmed by

local sources, or via translation, they were not included. An unintentional consequence of this is that a group that was prioritized/eligible/mandated to receive a COVID-19 vaccine may not have been captured by our dataset.

## Comparison to other data sources

We do not know of any other datasets publishing systematic data on COVID-19 vaccine policies. This unique source therefore adds considerable value to other datasets. For example, the Reuters COVID-19 vaccination tracker, although no longer updating data, reported the prioritization plan (similar to V1 - vaccine prioritization), availability and eligibility (similar to V2 - vaccination eligibility/availability) of different groups for vaccination in each country, although this does not provide the original archived source material, dates or a time series dataset to evidence when groups became eligible (https://www.reuters.com/graphics/world-coronavirus-tracker-and-maps/vaccination-rollout-and-access/). The Africa CDC Vaccine Dashboard also reports the date vaccination campaign commenced (similar to V2) in African nations, although it does not provide original archived source material, or state when which groups began to receive vaccines (https://africacdc.org/covid-19-vaccination/). Coronanet includes questions on the number of priority groups for vaccine distribution (similar to V1) but only through 2021 (https://www.coronanet-project.org/assets/CoronaNet_Codebook.pdf).

Vaccine policy data are particularly useful when combined with information on the number of vaccine doses administered. For example, the OxCGRT vaccine policy data are displayed as part of Our World in Data's COVID-19 vaccinations website resource[3], which combines country-specific vaccination data into a single resource. Our data can be compared to variables such as proportion of the population that has received one dose, vaccination uptake rates and number of doses administered, to consider the effects of vaccination policies on these. Our H7 indicator, which is a summary of our V2 vaccination policy indicator, is also displayed here. The data are also used by the United Nations Development Programme (UNDP) and WHO's Global Dashboard for Vaccine Equity (https://data.undp.org/vaccine-equity/), demonstrating their value in informing an interactive accessible tool to reflect the data on the COVID-19 rollout to ensure equitable access.

## Reporting summary

Further information on research design is available in the Nature Portfolio Reporting Summary linked to this article.

## Data availability

The vaccine policy data are available on GitHub (use the download tab to download linked file): https://github.com/OxCGRT/covid-policy-tracker/blob/master/data/OxCGRT_vaccines_full.csv All other information (including methodology and documentation) is available at https://github.com/OxCGRT.

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

## Acknowledgements

We thank the Oxford Martin School at the University of Oxford for funding the vaccination policy indicator design and collection; our research colleagues on the project, M. DiFolco, R. Nagesh, R. Barnes and M. Luciano; our database and software developer, T. Boby; our project manager, J. Elms; and our large team of dedicated volunteer data collectors who have committed thousands of hours to data collection and without whom we would not have been able to do it. A full list of volunteers is available at https://www.bsg.ox.ac.uk/research/covid-19-government-response-tracker. The funders had no role in study design, data collection and analysis, decision to publish or preparation of the manuscript.

## Author contributions

E.C.-B. and H.T. conceptualized the project. E.C.-B., H.T., K.G., H.Z., A. Pott, A.W. and T.B. curated the data. E.C.-B., H.T., B.A. and K.G. conducted formal data analysis. E.C.-B. and H.T. developed the methodology. E.C.-B., H.T. and B.A. provided resources. E.C.-B. and H.T. wrote the original draft. E.C.-B., H.T., B.A, A. Petherick, T.H. and T.P. edited the manuscript. A. Petherick, T.H. and T.P. supervised the project. All named authors read and approved the manuscript.

## Competing interests

The authors declare no competing interests.

## Additional information

**Correspondence and requests for materials** should be addressed to Emily Cameron-Blake.

# Reporting Summary

## Statistics

For all statistical analyses, confirm that the following items are present in the figure legend, table legend, main text, or Methods section.

| n/a | Confirmed | |
|---|---|---|
| ☐ | ☒ | The exact sample size (*n*) for each experimental group/condition, given as a discrete number and unit of measurement |
| ☐ | ☒ | A statement on whether measurements were taken from distinct samples or whether the same sample was measured repeatedly |
| ☒ | ☐ | The statistical test(s) used AND whether they are one- or two-sided *Only common tests should be described solely by name; describe more complex techniques in the Methods section.* |
| ☐ | ☒ | A description of all covariates tested |
| ☒ | ☐ | A description of any assumptions or corrections, such as tests of normality and adjustment for multiple comparisons |
| ☒ | ☐ | A full description of the statistical parameters including central tendency (e.g. means) or other basic estimates (e.g. regression coefficient) AND variation (e.g. standard deviation) or associated estimates of uncertainty (e.g. confidence intervals) |
| ☒ | ☐ | For null hypothesis testing, the test statistic (e.g. $F$, $t$, $r$) with confidence intervals, effect sizes, degrees of freedom and $P$ value noted *Give P values as exact values whenever suitable.* |
| ☒ | ☐ | For Bayesian analysis, information on the choice of priors and Markov chain Monte Carlo settings |
| ☒ | ☐ | For hierarchical and complex designs, identification of the appropriate level for tests and full reporting of outcomes |
| ☒ | ☐ | Estimates of effect sizes (e.g. Cohen's *d*, Pearson's *r*), indicating how they were calculated |

*Our web collection on statistics for biologists contains articles on many of the points above.*

## Software and code

Policy information about availability of computer code

| Data collection | Google translate (online web version, http://translate.google.com) was used in the data collection process to aid translation and understanding of data for collection. |
|---|---|
| Data analysis | Stata (V17.0), Microsoft Excel (V 16) and R (4.1.3) were used to manipulate and analyse the data for visualisations and Figures in this article. |

For manuscripts utilizing custom algorithms or software that are central to the research but not yet described in published literature, software must be made available to editors and reviewers. We strongly encourage code deposition in a community repository (e.g. GitHub). See the Nature Portfolio guidelines for submitting code & software for further information.

## Data

Policy information about availability of data

All manuscripts must include a data availability statement. This statement should provide the following information, where applicable:

- Accession codes, unique identifiers, or web links for publicly available datasets
- A description of any restrictions on data availability
- For clinical datasets or third party data, please ensure that the statement adheres to our policy

The vaccine policy data are available on GitHub (use the download tab to download linked file): https://github.com/OxCGRT/covid-policy-tracker/blob/master/data/OxCGRT_vaccines_full.csv . All other information (including methodology and documentation) is available at: https://github.com/OxCGRT

There are no restrictions on data availability.

## Human research participants

Policy information about [studies involving human research participants and Sex and Gender in Research.](studies involving human research participants and Sex and Gender in Research.)

| | |
|---|---|
| Reporting on sex and gender | N/A |
| Population characteristics | N/A |
| Recruitment | N/A |
| Ethics oversight | N/A |

Note that full information on the approval of the study protocol must also be provided in the manuscript.

## Field-specific reporting

Please select the one below that is the best fit for your research. If you are not sure, read the appropriate sections before making your selection.

☐ Life sciences    ☒ Behavioural & social sciences    ☐ Ecological, evolutionary & environmental sciences

For a reference copy of the document with all sections, see [nature.com/documents/nr-reporting-summary-flat.pdf](nature.com/documents/nr-reporting-summary-flat.pdf)

## Behavioural & social sciences study design

All studies must disclose on these points even when the disclosure is negative.

| | |
|---|---|
| Study description | This 'resource' article describes and presents a new dataset that provides quantitaive and qualitative data on COVID-19 vaccination policies around the globe. It does not carry out causal inference. |
| Research sample | The new data and database presented in this article includes information on most countries in the world (185), as well as a number of subnational jurisdictions in the United States, Canada, United Kingdom and China.  Countries were included in the dataset due to the ease/ability to collect the relevant data, and volunteer data collector capacity. The sample of countries in this dataset is representative of the global population and of each continent. |
| Sampling strategy | Sampling is not relied upon for this article. This article profiles a newly developed dataset of COVID-19 vaccine policies, and the countries therein were included based on the ability to find and record relevant policy data. Sufficient countries (185) are included in this dataset to enable cross-country comparisons. |
| Data collection | The OxCGRT database is maintained by a large team of specially trained volunteer data contributors from around the globe. Initially volunteers were recruited in March of 2020 largely from the postgraduate student body of the Blavatnik School of Government at the University of Oxford. Since then, additional contributors have been recruited through Oxford University departmental mailing lists, student societies and alumni lists as well as through social media channels of other large university networks and via a contributor group on LinkedIn. Many volunteers have been referred by existing or previous volunteers. To date, the OxCGRT has had over 1500 specially trained data-contributors.<br><br>New members of the data collection tam undergo a series of training steps. First, they complete a bespoke e-learning course (designed by lead author E.CB.) that describes which COVID-19 non-pharmaceutical intervention (NPI)  and vaccine policies the OxCGRT collects data on, how our ordinal/binary/categorical indicators are coded in the database, and how the database operates and how to contribute data. The e-learning course also includes practice questions and a test that must be passed with a minimum score of 80% in order to progress. The test assesses comprehension and understanding of the coding schema and collection process. After this training process, contributors are assigned to various 'teams' within the OxCGRT based on local knowledge, language, and needs of the program, and are assigned weekly tasks for updating data. All data collectors are expected to attend weekly contributor meetings where they are able to ask questions of the core OxCGRT team for clarity, if needed. For the vaccine policies, it was not possible to create an endless list of categories of people/groups, so a 'best-fit' table has been created and is available to both contributors and data users on GitHub.<br><br>OxCGRT collects national data on a weekly schedule, during which new task allocations are sent to the data collection team. For the addition of the vaccine policy dataset, experienced contributors were recruited to an initial team to begin building the data in the database from March 2021. Once each country/jurisdction was 'built' and updated with all vaccine policies, the allocation of updates to the vaccine policies was added into the regular weekly rotation of updating data for the OxCGRT. The data is published in real-time as contributors enter it into the system.<br><br>Once data is entered into the database, it is marked as 'provisional', which flags it for a review process. First, after each allocation round, a small team will do quick spot checks to ensure that data has been entered properly and there are no gross errors. The provisional data is then queued for attention by a more thorough review team. Initial reviews for this dataset were completed by |

lead authors E.CB. and H.T., then by specially trained reviewers on the vaccine policies (trained by a bespoke e-learning course). Initial reviews suggest a high degree of accuracy in the initial data collection (90% of initial entries were correct, the other 10% were corrected - often a simple date change or inclusion/exclusion of a particular group for vaccination).

Data is collected from publicly available sources such as government press releases and briefings, international organisation reports, and trusted news articles. Original source materials are archived and saved as notes within the database so that coding can be checked and substantiated.

For this article, there was no study hypotheis required/developed, nor a requirement to be blinded to experimental conditions.

| | |
|---|---|
| Timing | Collection of the vaccine policy data began in March 2021 and continues through the present. All data from January 1 2020 until 31 December 2022 are recorded in the database. |
| Data exclusions | This article contains no in-depth analysis, but presents 'snapshots' of the data to demonstrate potential uses and trends. No data are excluded from these presentations. |
| Non-participation | There are no participants in this study |
| Randomization | This study/article does not rely on randomization as it is presenting a new dataset, and some insights for motivation of future use of the dataset. |

# Reporting for specific materials, systems and methods

We require information from authors about some types of materials, experimental systems and methods used in many studies. Here, indicate whether each material, system or method listed is relevant to your study. If you are not sure if a list item applies to your research, read the appropriate section before selecting a response.

## Materials & experimental systems

| n/a | Involved in the study |
|---|---|
| ☒ | Antibodies |
| ☒ | Eukaryotic cell lines |
| ☒ | Palaeontology and archaeology |
| ☒ | Animals and other organisms |
| ☒ | Clinical data |
| ☒ | Dual use research of concern |

## Methods

| n/a | Involved in the study |
|---|---|
| ☒ | ChIP-seq |
| ☒ | Flow cytometry |
| ☒ | MRI-based neuroimaging |

