## [Peer Review File · Nature Human Behaviour]

Peer Review Information

Journal: Nature Human Behaviour

Manuscript Title: A panel dataset of COVID-19 vaccination policies in 185 countries

Corresponding author name(s): Emily Cameron-Blake

Editorial Notes:

Reviewer Comments & Decisions:

Decision Letter, initial version:

19th December 2022

Dear Ms Cameron-Blake,

Thank you once again for your manuscript, entitled "A global panel dataset of COVID-19 vaccination policies," and for your patience during the peer review process.

Your manuscript has now been evaluated by 3 reviewers, whose comments are included at the end of this letter. Although the reviewers find your work to be of interest, they also raise some important concerns. We are very interested in the possibility of publishing your study in Nature Human Behaviour, but would like to consider your response to these concerns in the form of a revised manuscript before we make a decision on publication.

To guide the scope of the revisions, the editors discuss the referee reports in detail within the team, including with the chief editor, with a view to (1) identifying key priorities that should be addressed in revision and (2) overruling referee requests that are deemed beyond the scope of the current study. We hope that you will find the prioritised set of referee points to be useful when revising your study. Please do not hesitate to get in touch if you would like to discuss these issues further.

1. The reviewers ask that you provide additional methodological information, such as how the database was created and how core indicators were selected and why. Please note that the journal's policies require the Method section to be included at the end of the article and we therefore ask you to not move the section to appear earlier in the main text. Instead, please provide some crucial methodological information in the main text, while providing more detailed information in the Method section.
2. Reviewer 2 asks that all conclusions presented in the manuscript are supported. Please either cite existing empirical work in support of your conclusions, or provide additional data and analyses that support them.
3. In your manuscript, please discuss additional variables of interest such as vaccine types and funding, as well as the value of the database beyond the COVID-19 context.

Finally, your revised manuscript must comply fully with our editorial policies and formatting requirements. Failure to do so will result in your manuscript being returned to you, which will delay its consideration. To assist you in this process, I have attached a checklist that lists all of our

2requirements. If you have any questions about any of our policies or formatting, please don't hesitate to contact me.

In sum, we invite you to revise your manuscript taking into account all reviewer and editor comments. We are committed to providing a fair and constructive peer-review process. Do not hesitate to contact us if there are specific requests from the reviewers that you believe are technically impossible or unlikely to yield a meaningful outcome.

We hope to receive your revised manuscript within two months. I would be grateful if you could contact us as soon as possible if you foresee difficulties with meeting this target resubmission date.

- Include a "Response to the editors and reviewers" document detailing, point-by-point, how you addressed each editor and referee comment. If no action was taken to address a point, you must provide a compelling argument. When formatting this document, please respond to each reviewer comment individually, including the full text of the reviewer comment verbatim followed by your response to the individual point. This response will be used by the editors to evaluate your revision and sent back to the reviewers along with the revised manuscript.
- Highlight all changes made to your manuscript or provide us with a version that tracks changes.

[REDACTED]

We look forward to seeing the revised manuscript and thank you for the opportunity to review your work. Please do not hesitate to contact me if you have any questions or would like to discuss these revisions further.

[REDACTED]

Reviewer expertise:

Reviewer #1: vaccine hesitancy, policy making in public health

Reviewer #2: public health, global health, COVID-19

Reviewer #3: public health in the Global South

REVIEWER COMMENTS:

Reviewer #1:

Remarks to the Author:

I appreciated the opportunity to review this article, which presents and describes features of a global panel dataset of COVID-19 vaccination policies collated since January 2020. The article has an interesting job to do: it is seeking to socialise and explain the dataset as well as to report some noteworthy findings from analysis of it. This is a challenging job to execute in the context of a straightforward article structure. In particular, I found myself wishing that the methods were included in the standard place (before the results) but it may be a journal requirement that they are at the end. This is a pity, though, because this is an article about how a global dataset was assembled and it would help us to know that as we go along. For example, at line 125 we are presented with 'economic function' as though that is self-evident, yet it is not clearly explained until later.

The authors have engaged in analysis of nine 'selected' eliminators and nine 'selected' mitigators. It matters how these were selected and why. Please tell us. (Again, it would be nice not to have to wait until the end to hear about this.)

At line 161 the authors suggest that LMICs and LICs may have provided universal vaccine access as an attempt to mitigate vaccine hesitancy. However, they may also have faced challenges reaching and accessing targeted groups, whereas making the vaccine available to everybody would be a blame avoidance strategy ('you can't say we didn't offer it to everybody.')

I appreciate that the authors are trying to use inclusive language when they refer to 'pregnant people' but in doing so it does not then make sense to say 'pregnant people are the only explicitly gendered group' since this terminology explicitly desexes women, whereas 'pregnant people' would necessarily be of either gender. You may avoid this by considering pregnancy instead as a subset of comorbidity, albeit a unique and important one – or refer to pregnant women, who can then be conceived of as a sex class.

Line 226 – would love to hear more about vaccine tourism. From where to where? How was this promoted / encouraged?

When we finally get to the methods, they are clear and well-executed, containing all the details we could want (e.g. date of announcement versus date of enforcement, clear definition of what is and is not a mandate, staying out of questions of enforcement.) I was left wondering how the issue of translation was tackled – were the volunteer researchers bilingual in-country researchers? Was translation software ever used?

The dataset sounds extremely useful and appealing. The article is an excellent way of socialising that dataset. It would be great if the editors can advise the authors whether they might include the methods prior to the results.

Reviewer #2:

Remarks to the Author:

3The authors have created a database for tracking COVID-19 vaccination policies. In this manuscript, they describe the database and indicators used and provide examples as to how the database can be used. It is great to have this organized database and especially important that it is made publicly available.

Major comments:

Given that this manuscript is meant to serve as a descriptive piece that introduces the readers to the database, I expected to read more information as to how the database was created. How were the indicators chosen and who verified the information? Some of this comes out in the limitations but the description of the creation of the database is lacking in the Methods section. Some of this comes later in the paper but I think it would be better to include it earlier before showing the examples.

In several instances throughout the Results, the authors draw conclusions about variations in countries that seem lacking in analysis. Two examples that particularly stand out:

On page 4, lines 114-116 the authors state: "Many countries in the WHO West Pacific Region adopted policies closer to the elimination model, perhaps in part due to the region's prior experience with SARS." Are there citations that could be included to strengthen this idea? Could it also be plausible that island nations are more likely to try and adopt elimination models that landlocked countries that more easily share borders?

On page 6, the authors propose that vaccine hesitancy may explain why LMIC and LIC countries were more likely to swiftly move to universal access to vaccinations. I also wonder if it this trend can also be explained by the fact that these countries received vaccines/rolled out vaccinations later than HIC countries and thus moved to universal access at the same time as the "global trend" was also providing universal access?

Minor comments:

What about types of vaccines? What about internal vs external funding for vaccines?

What is the value of this database beyond COVID-19? Some discussion of this may be an interesting idea for thinking about future pandemics or additional vaccine rollout initiatives.

Reviewer #3:

Remarks to the Author:

Written text can be shortened to avoid repetition.

Format of the manuscript such as Title, Abstract, Keyword, Introduction, Methods, Results, Discussion, limitations, conclusion, recommendations, references. Tables and figures are not very effective in presenting the data.

Author Rebuttal to Initial comments

Reviewer 1	Author Response
I appreciated the opportunity to review this article, which presents and describes features of a global panel dataset of COVID-19 vaccination policies collated since January 2020. The article has an interesting job to do: it is seeking to socialise and explain the dataset as well as to report some noteworthy findings from analysis of it. This is a challenging job to execute in the context of a straightforward article structure. In particular, I found myself wishing that the methods were included in the standard place (before the results) but it may be a journal requirement that they are at the end. This is a pity, though, because this is an article about how a global dataset was assembled and it would help us to know that as we go along. For example, at line 125 we are presented with ‘economic function’ as though that is self-evident, yet it is not clearly explained until later.	Thank you for this thoughtful commentary and appreciation for the specific goals of this type of article. We have added some additional text to the introductory paragraphs to provide a sufficient sense of the data collection before the methods section arrives. We have also added some additional text to the methods on how the dataset was developed. With direction from the editors, we are unable to deviate from the set structure of resource papers, so moving methods to the front is not possible. But hopefully the additional text (especially around terms such as ‘economic function’) aids the reader prior to the methods section later on. Lines 65-68, 76-79, 375-385
The authors have engaged in analysis of nine ‘selected’ eliminators and nine ‘selected’ mitigators. It matters how these were selected and why. Please tell us. (Again, it would be nice not to have to wait until the end to hear about this.)	These nine ‘selected’ eliminators and mitigators (Table 2) were selected as examples from an extensive list of eliminators, see supplementary table S1) We have also deepened our analyses of all countries that were characterized as adopting elimination and mitigation strategies. According to related literature, we have adopted three definitions:  • Because many countries in the WHO West Pacific Region adopted policies closer to the elimination model, perhaps in part due to the region’s prior experience with SARS, we have used WHO-PR membership as a variable. • Relatedly, it has also been hypothesised that island nations may have been more likely to pursue elimination models during COVID-19 due to their geographical advantage of

	isolation and ability to seal national borders, though to many LICs/LMICs this came at significant economic cost. We thus compare island vs. non-island nations. • Finally, early responses to the Covid-19 pandemic have also brought a distinction between mitigation and elimination strategies based on the OECD countries (BMJ, 2020). We have also used this metric. We have inserted references for each of these country groupings and added a further description in the manuscript. Lines: 129-178 We have added a comprehensive list of eliminators and mitigators (defined by the WHO-PR, OECD, and island nations) as a table to the supplementary materials (Supplementary Table S1).
At line 161 the authors suggest that LMICs and LICS may have provided universal vaccine access as an attempt to mitigate vaccine hesitancy. However, they may also have faced challenges reaching and accessing targeted groups, whereas making the vaccine available to everybody would be a blame avoidance strategy ('you can't say we didn't offer it to everybody.')	Thank you for this. We have simply provided an example of a possible reason for this swift move, rather than a speculation as to what may have occurred. Further research and discussions with the various governments of the LMICs and LICs would be necessary to determine the actual reasons. However, we have added another possible explanation as suggested, being that it could have been an attempt to avoid public perception of favoring or excluding certain groups in the face of a health emergency and low supply of vaccines and appearing to be providing equal access for all residents, despite the challenges that may present socially and geographically.

	Lines 209-214
I appreciate that the authors are trying to use inclusive language when they refer to ‘pregnant people’ but in doing so it does not then make sense to say ‘pregnant people are the only explicitly gendered group’ since this terminology explicitly desexes women, whereas ‘pregnant people’ would necessarily be of either gender. You may avoid this by considering pregnancy instead as a subset of comorbidity, albeit a unique and important one – or refer to pregnant women, who can then be conceived of as a sex class.	We have edited the text to address this point. Lines 241-247
Line 226 – would love to hear more about vaccine tourism. From where to where? How was this promoted / encouraged?	Thank you for this comment. We have added more context and reference/media links to elaborate on this issue. Lines 288-295
When we finally get to the methods, they are clear and well-executed, containing all the details we could want (e.g. date of announcement versus date of enforcement, clear definition of what is and is not a mandate, staying out of questions of enforcement.) I was left wondering how the issue of translation was tackled – were the volunteer researchers bilingual in-country researchers? Was translation software ever used?	Thanks for pointing this out. We have added a line to the “data collection” section in methods on this point. Our pool of OxCGRT volunteer data contributors consisted of 90+ unique languages spoken, and where we struggled to find English sources, or sources that were not well translated into English, we consulted this international pool of volunteers. To your question about translation software, we recommended that our volunteers use free software and programs such as Google Translate - this was particularly helpful when looking for the dates of policy effect and groups prioritised/eligible. We have also made a note of this in our methods. As also noted in the methods, all notes in the database are recorded

	in English, whether the source material was English or not. Lines: 442- 448
Reviewer 2	
Given that this manuscript is meant to serve as a descriptive piece that introduces the readers to the database, I expected to read more information as to how the database was created. How were the indicators chosen and who verified the information? Some of this comes out in the limitations but the description of the creation of the database is lacking in the Methods section. Some of this comes later in the paper but I think it would be better to include it earlier before showing the examples.	Thank you for this comment. We have added a bit more context to the section on “Indicators” in the methods section, and also alluded to this in the intro section to the paper. Based on extensive consultations with governments, policy makers around the world, experts in the field and the OxCGRT expert advisory board, we considered many different options of policies to collect, but felt that at the time these 4 policies were lacking in existing data sets, and potentially critical to informing rollouts for future pandemics. We also considered our volunteer capacity as they would need to be trained and monitored. We have also added additional information about who collected/verified data in the methods section. Lines 375- 385, 442-459, 480-498 Due to the limitations of the NHB Resource format, we are unable to move this section to earlier in the manuscript.
In several instances throughout the Results, the authors draw conclusions about variations in countries that seem lacking in analysis. Two examples that	Thank you. We have also addressed this above, as a comment from Reviewer 1.

particularly stand out: On page 4, lines 114-116 the authors state: “Many countries in the WHO West Pacific Region adopted policies closer to the elimination model, perhaps in part due to the region’s prior experience with SARS.” Are there citations that could be included to strengthen this idea? Could it also be plausible that island nations are more likely to try and adopt elimination models that landlocked countries that more easily share borders?	It is certainly beyond the scope of this resource piece to explain the descriptive patterns identified, but we seek to note where patterns relate to claims others have made in the literature in order to highlight opportunities for further research. To that end, we have added citations to this conjecture, which several authors have made. We have also added a line pertaining to the idea that island nations may have been more likely to pursue elimination models due to their geographical advantage of isolation and sealing national borders, with cited studies that put forward this proposition: Lines 129 - 168
On page 6, the authors propose that vaccine hesitancy may explain why LMIC and LIC countries were more likely to swiftly move to universal access to vaccinations. I also wonder if it this trend can also be explained by the fact that these countries received vaccines/rolled out vaccinations later than HIC countries and thus moved to universal access at the same time as the “global trend” was also providing universal access?	We have also responded to a comment from Reviewer 1 on this issue. We have purposefully avoided insinuating any global trends, rather we are presenting some patterns and potential trends. Further analysis using this dataset might prove whether such trends did indeed exist. However, we have adjusted the text, and added another possible explanation for the swift move to universal vaccination, supported by additional references.

	Lines 209-214.
What about types of vaccines? What about internal vs external funding for vaccines?	These are very relevant information points that would be valuable to collect, and indeed we had identified them as potential candidates for inclusion. However, because of capacity constraints these were not prioritized (see rationale above). These would indeed be helpful to consider in future studies and may be an ambition for expansion. We have elaborated on these briefly in lines 380-384. Incidentally, in a related project with UNDP we were able to explore further indicators in a less systematic fashion, including rural vs urban distribution, vaccine numbers acquired by countries, brands acquired, and whether the vaccines were funded by the government or provided by COVAX. We have mentioned the UNDP dashboard in the manuscript, as a complementary data set/source. (Line 204) https://data.undp.org/vaccine-equity/
What is the value of this database beyond COVID-19? Some discussion of this may be an interesting idea for thinking about future pandemics or additional vaccine rollout initiatives.	While policies around the rollout of COVID-19 vaccines may not apply to all vaccines, we propose that researchers and policymakers seeking to prepare for future pandemics will be very keen to understand the relative effects and effectiveness of different policy choices around the COVID-19 pandemic. Comparison between countries with similar demographics, or those with unique demographics and geography might highlight best practice for future pandemics. More speculatively, the salience and, in some context, controversies around COVID-19 vaccines may have effects that shape vaccine policies more generally. Either way, careful understanding of COVID-19 vaccine policies is likely to be relevant for a wide array of

	future public health strategies involving vaccination. We have added a sentence at the end of the introduction to note this possibility. Lines 101 - 106 Note: the information gathered was delivered and utilized in real-time by the UK Government and by the UNDP for their Global Vaccine Equity Dashboard.
Reviewer 3	
Written text can be shortened to avoid repetition. Format of the manuscript such as Title, Abstract, Keyword, Introduction, Methods, Results, Discussion, limitations, conclusion, recommendations, references. Tables and figures are not very effective in presenting the data.	Following discussion with the editor, no actions are needed. There are set formats for Resource papers.

Decision Letter, first revision:

Our ref: NATHUMBEHAV-22102653A

30th March 2023

Dear Dr. Cameron-Blake,

Thank you for submitting your revised manuscript "A Global Panel dataset of COVID-19 Vaccination Policies" (NATHUMBEHAV-22102653A). It has now been seen by the original referees and their

11comments are below. As you can see, the reviewers find that the paper has improved in revision. We will therefore be happy in principle to publish it in Nature Human Behaviour, pending minor revisions to comply with our editorial and formatting guidelines.

We are now performing detailed checks on your paper and will send you a checklist detailing our editorial and formatting requirements within a week. Please do not upload the final materials and make any revisions until you receive this additional information from us.

[REDACTED]

Reviewer #1 (Remarks to the Author):

I'm very happy with the way that the authors revised this paper in response to reviewers. It's great and I hope to see it published soon.

Reviewer #2 (Remarks to the Author):

Thank you for the thoughtful consideration and responses to my earlier comments. I have no further comments.

Our ref: NATHUMBEHAV-22102653A

5th April 2023

Dear Dr. Cameron-Blake,

Thank you for your patience as we've prepared the guidelines for final submission of your Nature Human Behaviour manuscript, "A Global Panel dataset of COVID-19 Vaccination Policies" (NATHUMBEHAV-22102653A). Please carefully follow the step-by-step instructions provided in the attached file, and add a response in each row of the table to indicate the changes that you have made. Please also check and comment on any additional marked-up edits we have proposed within the text. Ensuring that each point is addressed will help to ensure that your revised manuscript can be swiftly handed over to our production team.

12We would hope to receive your revised paper, with all of the requested files and forms within two-three weeks. Please get in contact with us if you anticipate delays.

Nature Human Behaviour offers a Transparent Peer Review option for new original research manuscripts submitted after December 1st, 2019. As part of this initiative, we encourage our authors to support increased transparency into the peer review process by agreeing to have the reviewer comments, author rebuttal letters, and editorial decision letters published as a Supplementary item. When you submit your final files please clearly state in your cover letter whether or not you would like to participate in this initiative. Please note that failure to state your preference will result in delays in accepting your manuscript for publication.

In recognition of the time and expertise our reviewers provide to Nature Human Behaviour's editorial process, we would like to formally acknowledge their contribution to the external peer review of your manuscript entitled "A Global Panel dataset of COVID-19 Vaccination Policies". For those reviewers who give their assent, we will be publishing their names alongside the published article.

Cover suggestions

As you prepare your final files we encourage you to consider whether you have any images or illustrations that may be appropriate for use on the cover of Nature Human Behaviour.

ORCID

Non-corresponding authors do not have to link their ORCID IDs but are encouraged to do so. Please note that it will not be possible to add/modify ORCID IDs at proof. Thus, please let your co-authors know that if they wish to have their ORCID added to the paper they must follow the procedure described in the following link prior to acceptance: <https://www.springernature.com/gp/researchers/orcid/orcid-for-nature-research>

Nature Human Behaviour has now transitioned to a unified Rights Collection system which will allow our Author Services team to quickly and easily collect the rights and permissions required to publish your work. Approximately 10 days after your paper is formally accepted, you will receive an email in providing you with a link to complete the grant of rights. If your paper is eligible for Open Access, our Author Services team will also be in touch regarding any additional information that may be required to arrange payment for your article. Please note that you will not receive your proofs until the publishing agreement has been received through our system.

Please note that *Nature Human Behaviour* is a Transformative Journal (TJ). Authors may publish their research with us through the traditional subscription access route or make their paper immediately open access through payment of an article-processing charge (APC). Authors will not be required to make a final decision about access to their article until it has been accepted. [Find out more about Transformative Journals](https://www.springernature.com/gp/open-research/transformative-journals)

Authors may need to take specific actions to achieve [compliance with funder and institutional open access mandates](https://www.springernature.com/gp/open-research/funding/policy-compliance-faqs). If your research is supported by a funder that requires immediate open access (e.g. according to [Plan S principles](https://www.springernature.com/gp/open-research/plan-s-compliance)) then you should select the gold OA route, and we will direct you to the compliant route where possible. For authors selecting the subscription publication route, the journal's standard licensing terms will need to be accepted, including [self-archiving policies](https://www.springernature.com/gp/open-research/policies/journal-policies). Those licensing terms will supersede any other terms that the author or any third party may assert apply to any version of the manuscript.

Please use the following link for uploading these materials:
[REDACTED]

[REDACTED]

Reviewer #1:

Remarks to the Author:

I'm very happy with the way that the authors revised this paper in response to reviewers. It's great and I hope to see it published soon.

Reviewer #2:

Remarks to the Author:

Thank you for the thoughtful consideration and responses to my earlier comments. I have no further comments.

Final Decision Letter:

Dear Ms Cameron-Blake,

We are pleased to inform you that your Resource "A panel dataset of COVID-19 vaccination policies in 185 countries", has now been accepted for publication in Nature Human Behaviour.

Please note that *Nature Human Behaviour* is a Transformative Journal (TJ). Authors whose manuscript was submitted on or after January 1st, 2021, may publish their research with us through the traditional subscription access route or make their paper immediately open access through payment of an article-processing charge (APC). Authors will not be required to make a final decision about access to their article until it has been accepted. IMPORTANT NOTE: Articles submitted before January 1st, 2021, are not eligible for Open Access publication. [Find out more about Transformative Journals](https://www.springernature.com/gp/open-research/transformative-journals)

Authors may need to take specific actions to achieve [compliance with funder and institutional open access mandates](https://www.springernature.com/gp/open-research/funding/policy-compliance-faqs). If your research is supported by a funder that requires immediate open access (e.g. according to [Plan S principles](https://www.springernature.com/gp/open-research/plan-s-compliance)) then you should select the gold OA route, and we will direct you to the compliant route where possible. For authors selecting the subscription publication route, the journal's standard licensing terms will need to be accepted, including [self-archiving policies](https://www.springernature.com/gp/open-research/policies/journal-policies). Those licensing terms will supersede any other terms that the author or any third party may assert apply to any version of the manuscript.

Once your manuscript is typeset and you have completed the appropriate grant of rights, you will receive a link to your electronic proof via email with a request to make any corrections within 48 hours. If, when you receive your proof, you cannot meet this deadline, please inform us at rjsproduction@springernature.com immediately. Once your paper has been scheduled for online

15publication, the Nature press office will be in touch to confirm the details.

[REDACTED]

P.S. Click on the following link if you would like to recommend Nature Human Behaviour to your librarian <http://www.nature.com/subscriptions/recommend.html#forms>

** Visit the Springer Nature Editorial and Publishing website at http://editorial-jobs.springernature.com?utm_source=ejp_NHumB_email&utm_medium=ejp_NHumB_email&utm_campaign=ejp_NHumB for more information about our career opportunities. If you have any questions please click [here](mailto:editorial.publishing.jobs@springernature.com). **